# INTERPRETABLE NEUROPSYCHIATRIC DIAGNOSIS VIA CONCEPT-GUIDED GRAPH NEURAL NETWORKS

## ABSTRACT

Nearly one in five adolescents currently live with a diagnosed mental or behavioral health condition, such as anxiety, depression, or conduct disorder, underscoring the urgency of developing accurate and interpretable diagnostic tools. Resting-state functional magnetic resonance imaging (rs-fMRI) provides a powerful lens into large-scale functional connectivity, where brain regions are modeled as nodes and inter-regional synchrony as edges, offering clinically relevant biomarkers for psychiatric disorders. While prior works use graph neural network (GNN) approaches for disorder prediction, they remain complex black-boxes, limiting their reliability and clinical translation. In this work, we propose CONCEPTNEURO, a concept-based diagnosis framework that leverages large language models (LLMs) and neurobiological domain knowledge to automatically generate, filter, and encode interpretable functional connectivity concepts. Each concept is represented as a structured subgraph linking specific brain regions, which are then passed through a concept classifier. Our design ensures predictions through clinically meaningful connectivity patterns, enabling both interpretability and strong predictive performance. Extensive experiments across multiple psychiatric disorder datasets demonstrate that CONCEPTNEURO-augmented GNNs consistently outperform their vanilla counterparts, improving accuracy while providing transparent, clinically aligned explanations. Furthermore, concept analyses highlight disorder-specific connectivity patterns that align with expert knowledge and suggest new hypotheses for future investigation, establishing CONCEPTNEURO as an interpretable, domain-informed framework for psychiatric disorder diagnosis.

## 1 INTRODUCTION

Functional magnetic resonance imaging (fMRI) is a powerful non-invasive tool that captures dynamic changes in neural activity through blood-oxygen-level-dependent (BOLD) responses (Fox & Raichle, 2007). Functional connectivity (FC), defined as the degree of similarity in BOLD activity across brain regions–particularly during resting-state–has emerged as an important tool to quantify individual differences in cognition (Bassett & Sporns, 2017) and behavior (Finn et al., 2015; Shen et al., 2017), as well as for diagnosing and characterizing brain disorders (Jo et al., 2019; Eslami et al., 2019). However, due to the high dimensionality of FC and the lack of domain-specific constraints, current predictive efforts based on this signal typically yield unreliable models that fail to translate into clinical practice and produce findings with limited interpretability.

To this end, there are two key obstacles in advancing FC-based neuropsychiatric diagnosis. First, existing models fail to integrate existing domain knowledge into the modeling framework. Neuroimaging and psychiatric research already provide rich knowledge about disorder-related regions and networks, yet most learning frameworks treat FC graphs as unstructured input, leaving this information unused and resulting in models that may overfit noise or miss clinically relevant signals. Second, current predictive models lack interpretability, *i.e.,* black-box approaches such as standard graph neural networks (GNNs) may achieve competitive accuracy but provide little insight into disorder-specific connectivity patterns, limiting clinical trust and scientific discovery.

**Present work.** To address these challenges, we propose CONCEPTNEURO, a novel framework that integrates large language models (LLMs) with graph-based concept modeling. Our framework is formulated using concept bottleneck models (CBMs) (Koh et al., 2020; Yeh et al., 2020) that

implicitly introduce an intermediate layer of human-understandable concepts between raw data representations and downstream predictions. Under the CBM framework, our approach tackles both obstacles in FC-based neuropsychiatric diagnosis with two novel designs: (1) **LLM-guided concept generation.** To incorporate prior neurobiological knowledge, we collect disorder-related terms from established neuroimaging resources (*e.g.*, NeuroQuery (Dockès et al., 2020)), which serve as anchors for generating candidate concepts. Guided by these terms, LLMs automatically produce diverse and disorder-specific functional connectivity concepts that capture clinically meaningful interactions between brain regions. (2) **Connectivity-based concept modeling.** To ensure interpretability, we introduce a concept bottleneck layer that mediates predictions through human-understandable connectivity concepts. Specifically, each concept is represented as a relationship between two groups of brain regions, where the connectivity strength is reflected by the edges in the induced subgraph. These subgraphs are then structurally encoded and passed into a concept classifier for disorder prediction. With these designs, CONCEPTNEURO is, to our knowledge, the first framework to enable automated concept generation for fMRI-based connectivity analysis. By bridging raw FC data with clinically interpretable representations, our method delivers accurate classification of psychiatric disorders while providing transparent, disorder-specific explanations that align with neurobiological insights.

We validate our framework on the task of disorder diagnosis, using fMRI datasets associated with multiple psychiatric disorders (*e.g.*, anxiety or conduct disorder). Experimental results demonstrate that CONCEPTNEURO achieves improved prediction accuracy and inherent interpretability compared to standard black-box approaches. In particular, the selected concepts highlight brain region interactions that align with knowledge of domain experts. The contributions of this work can be summarized as follows: ① we introduce an automated pipeline for generating and filtering clinically meaningful connectivity-based concepts from brain fMRI data; ② we design a concept-based GNN that integrates these concepts into the classification of brain disorders; and ③ we conduct comprehensive experiments showing that our framework balances predictive performance with interpretability, offering insights into disorder-specific brain connectivity patterns.

## 2 RELATED WORK

This work lies at the intersection of brain network analysis and concept-guided graph neural networks. Below, we discuss related work for each of these topics.

**Brain Network Analysis.** Analyzing brain networks focuses on uncovering the complex patterns of connectivity in the human brain (Cui et al., 2022; Kan et al., 2022; Zhang et al., 2022b). This line of research has recently attracted significant attention due to its wide range of applications, such as detecting biomarkers for neurological disorders (Yang et al., 2022), providing insights into cognitive mechanisms (Liu et al., 2023; Chen et al., 2024), and characterizing differences across various brain network types (Liao et al., 2024). A central task in this domain is predicting brain-related attributes, including demographic information and cognitive or task-specific states (Said et al., 2023; He et al., 2020). Graph neural networks (GNNs) have emerged as a dominant methodology for these prediction tasks (Li et al., 2022; Cui et al., 2022), owing to their strength in modeling structured relational data (Li et al., 2021; Xu et al., 2024; Wang et al., 2022).

**Concept-guided Graph Neural Networks.** Concept bottleneck models (CBMs) enhance interpretability by mediating predictions through human-understandable concepts (Koh et al., 2020; Yeh et al., 2020). They have been applied in domains such as radiology and pathology to link predictions with clinically meaningful features (Kim et al., 2018; Sauter et al., 2022), though most approaches depend on manually defined concept sets, limiting scalability. Recent advances leverage large language models (LLMs) to automatically propose candidate concepts (Kim et al., 2023; Yang et al., 2023), reducing annotation costs while maintaining interpretability. In graph learning, explainability has traditionally relied on post-hoc methods that highlight predictive subgraphs (Xuanyuan et al., 2023; Huang et al., 2022), but these offer only local, instance-level insights. To move toward global interpretability, recent work explores embedding concepts into GNN architectures. For example, neuron-level analyses reveal that hidden units can act as detectors of interpretable graph motifs (Xu et al., 2024), while Graph Concept Bottleneck Models explicitly encode concept relationships as a graph structure to improve transparency (Xu et al., 2025). Despite these advances, concept-guided GNNs remain underexplored, with most studies focusing on vision or tabular data rather than

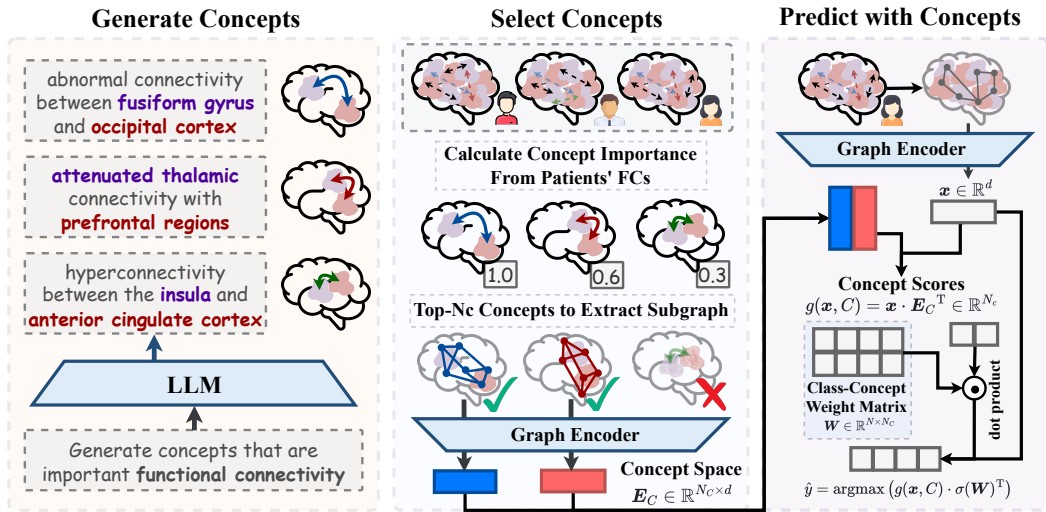

Figure 1: **Overview of the Proposed Framework. (Left)** We first prompt LLMs to generate disorder-specific functional connectivity concepts, which are refined through filtering to remove irrelevant or redundant concepts, resulting in a compact set of $N_c$ concepts. **(Center)** For each subject, we extract subgraphs corresponding to these concepts and encode them together with the subject's input functional connectivity graph. **(Right)** Finally, we compute concept scores, which are then passed through a concept bottleneck classifier to perform disorder prediction in an interpretable manner.

graph-structured domains. This motivates frameworks that can automatically generate and integrate domain-specific connectivity concepts for tasks such as neuroimaging-based diagnosis.

## 3 CONCEPTNEURO: OUR FRAMEWORK

In this section, we describe our framework that aims to perform neuropsychiatric diagnosis using interpretable connectivity concepts. To achieve this goal, CONCEPTNEURO leverages concept generation using LLMs (Sec. 3.2), connectivity-based concept modeling (Sec. 3.3), and training a concept classifier for neuropsychiatric diagnosis (Sec. 3.4). In Fig. 1, we present an overview of our end-to-end framework.

### 3.1 PRELIMINARIES

**Problem Formulation.** Let $\mathcal{D} = (G_i, y_i)_{i=1}^M$ be the dataset containing $M$ subject samples, where $G_i = (V, E, A_i)$ is the fMRI graph for subject $i$ and $y_i \in \mathcal{Y}$ is the disorder label from $N = |\mathcal{Y}|$ classes. In our task, $N$ could be two for binary classification or greater than two for the identification of multiple disorders. Notably, we work on a specific parcellation $V$ (*e.g.,* 100–400 ROIs, depending on the atlas) for each dataset. The subject-specific weighted adjacency $A_i \in \mathbb{R}^{|V| \times |V|}$ is derived from ROI time series (details in Sec. 3.1) obtained based on Pearson's correlation. Let $\mathcal{S} = \{c_1, \ldots, c_{N_C}\}$ denote the set of candidate concepts, where concept $c$ is defined as a pair of disjoint region sets $(V_c^1, V_c^2)$ with $(V_c^1, V_c^2) \subseteq V$, accompanied by a *direction prior* $\delta_c \in \{-1, +1\}$ indicating hypothesized (negative/positive) connectivity.

**From fMRI to Graphs.** We follow the minimal preprocessing pipeline for fMRI data (Glasser et al., 2013; Hagler et al., 2019), which includes motion correction, B0 distortion correction, and gradient nonlinearities distortion correction. To further minimize motion-related confounds, we apply iterative spatial smoothing, regression of motion parameters, and frame censoring (Qu et al., 2025; Yu et al., 2025). After preprocessing, ROI-level fMRI time-series were calculated as the mean of the values of the voxels within the ROI from the selected parcellation scheme. Formally, for each subject $i$, ROI $v$, we demote the extracted time series as $x_i(v) \in \mathbb{R}^T$, where $T$ is the number of time points.

## 3.2 Concept Generation using Large Language Models

A key challenge in building interpretable concept-based models is generating a sufficiently rich yet clinically meaningful set of candidate concepts. To this end, we design structured prompts for LLMs that guide them to propose *connectivity concepts* directly tied to functional brain regions. We design the prompts to ask to elicit short, specific features of fMRI connectivity that distinguish a given disorder. Meanwhile, we explicitly instruct the model to avoid features that are not based on functional connectivity (*e.g.,* fractional anisotropy measures) and to restrict outputs to pairs or sets of well-defined brain regions anchored in prior neuroimaging knowledge. Importantly, we collect disorder-related terms from established neuroimaging resources (*e.g.,* NeuroQuery (Dockès et al., 2020)) to provide additional guidance for the LLMs. For instance, the related terms for anxiety may include *amygdala*, *prefrontal*, *insula*, and *cingulate*.

Each generated concept follows the pattern of concise phrases such as "*reduced connectivity between amygdala and dorsolateral prefrontal cortex*" or "*hyperconnectivity between posterior cingulate cortex and medial prefrontal cortex.*" These are then parsed into structured region sets $(V_c^1, V_c^2)$ using atlas-specific aliases and ontology lookups, ensuring compatibility with downstream graph-based analysis. The motivation behind this design is twofold: i) leveraging LLMs' ability to synthesize domain knowledge into a broad and diverse candidate pool and ii) enforcing domain-informed constraints so that the resulting concepts remain interpretable, clinically relevant, and directly mappable onto fMRI connectivity data.

> **Example prompt template to generate concepts for 'Anxiety'**
>
> You need to list the most important visual features of brain images for diagnosing a patient as "Anxiety". You should be specific in generating these features that are related to single regions. You should make each feature very concise and clear, and each feature should be separated by a new line. You should not include any other information or explanation, just the features.
> You should make sure the generated concepts are not fractional anisotropy (FA) measures.
> You should make sure the generated concepts are derived only to functional connectivity.
> The generated concepts are related to at least one of the following: [Anxiety-related Terms]

To refine the LLM outputs, we apply rule-based filters: we discard concepts that involve fewer than two regions in either set, overlap excessively with existing concepts, or cannot be reliably resolved to atlas-defined regions. The resulting set $\mathcal{S}$ forms a compact yet diverse collection of connectivity-based concepts, which balances coverage across disorders while avoiding redundancy.

**Illustrative Example.** A concept "*hyperconnectivity between amygdala and prefrontal cortex*" maps to $V_c^1 = $ '*L/R amygdala*' and $V_c^2 = $ '*L/R ofrontal cortex parcels*'. For each subject $i$, we will examine only the edges between these sets to compute a *concept score* and a *subgraph embedding*.

## 3.3 Structural Encoding for Concept Subgraphs

With the concepts obtained from the LLMs' output for each disorder, we need to extract the subgraph of each concept to be integrated into our concept-guided GNN. Particularly, for a concept $c \in \mathcal{S}$ and subject $i$, the concept subgraph is given by:

$$G_i^c = \left(V_c^1 \cup V_c^2, \ E_c, \ A_i^c\right), \quad E_c = \{(u,v) \mid u \in V_c^1, \ v \in V_c^2\}, \tag{1}$$

with $A_i^c$ the $|V_c^1| \times |V_c^2|$ submatrix of $A_i$. Based on the extracted concept subgraph, we select $N_c$ concepts based on their average adjacency strength across subjects. Formally, for a concept $c \in \mathcal{S}$ with subgraph adjacency $A_i^c$ for subject $i$, we define its average connectivity score as

$$\bar{s}_c = \frac{1}{M} \sum_{i=1}^{M} \frac{1}{|E_c|} \sum_{(u,v) \in E_c} A_i^c(u,v), \tag{2}$$

where $M$ is the number of subjects and $E_c$ is the edge set of the concept subgraph. We then rank all candidate concepts $\{c\}$ by $\bar{s}_c$ and retain the top $N_c$ concepts to form the final concept set.

To capture higher-order structural patterns within each concept subgraph, we construct initial node features using two components: a one-hot vector indicating the ROI identity in the atlas, and the *average neighbor degree* of the node, computed with `networkx`, which summarizes the typical

degree of its neighbors and thus provides a local connectivity descriptor. Given these initial features, we learn structural embeddings of nodes through an edge-weighted message-passing framework. Formally, the node representation at layer $t$ is updated as:

$$\mathbf{h}_u^{(0)} = \left[ \mathbf{e}_u \; \middle\| \; \frac{1}{\deg(u)} \sum_{v \in \mathcal{N}(u)} \deg(v) \right], \tag{3}$$

where $\mathbf{e}_u \in \{0,1\}^{|V|}$ is the one-hot vector corresponding to ROI $u$, $\deg(u)$ is the degree of node $u$, $\mathcal{N}(u)$ is the neighbor set of $u$, and $[\cdot \, \| \, \cdot]$ denotes concatenation of feature vectors.

**General message passing.** For node $u$ at layer $t \rightarrow t+1$, the encoding process is described as follows:

$$\mathbf{h}_u^{(t+1)} = U^{(t)}\Big(\mathbf{h}_u^{(t)}, \bigoplus_{v \in \mathcal{N}(u)} M^{(t)}\big(\mathbf{h}_u^{(t)}, \mathbf{h}_v^{(t)}, A_i(u,v)\big)\Big), \tag{4}$$

where $\mathbf{h}_u^{(t)} \in \mathbb{R}^{d_t}$ is the node state, $\mathcal{N}(u)$ is the neighborhood of $u$, $\bigoplus \in \{\sum, \mathrm{mean}, \mathrm{max}\}$ is a permutation-invariant aggregator, $M^{(t)}$ is a learnable message function, $U^{(t)}$ is a learnable update function. After $T$ layers, we obtain the final subgraph embedding via attention pooling:

$$\alpha_u = \mathrm{softmax}\Big(w^\top x_u^{(T)}\Big), \quad \mathbf{h}_i^c = \sum_{u \in V_c^1 \cup V_c^2} \alpha_u \mathbf{h}_u^{(T)}. \tag{5}$$

This design ensures that both ROI identity and local structural properties contribute to the learned representation, while message passing integrates higher-order connectivity patterns across regions. We denote the structural encoder by $g(\cdot)$, and obtain the concept-specific representation of subject $i$ as $\mathbf{h}_i^c = g(G_i^c)$, where $G_i^c$ is the concept subgraph for subject $i$.

## 3.4 CONCEPT BOTTLENECK CLASSIFIER

For each subject input $G_i$, we obtain its embedding $\mathbf{z}_i = g(G_i)$ using the structural encoder described in Sec. 3.3. We then compute its concept score $s_{i,c}$ for each concept $c$ via dot product:

$$s_{i,c} = \mathbf{z}_i^\top \mathbf{h}_i^c, \quad \forall c \in \mathcal{S}. \tag{6}$$

This yields a concept similarity vector $\mathbf{s}_i = [s_{i,1}, s_{i,2}, \ldots, s_{i,N_c}] \in \mathbb{R}^{N_c}$, where $N_c$ is the number of concepts. The vector $s_i$ serves as the concept-level representation of subject $i$, which is then passed through a multi-layer perceptron (MLP) classifier to produce the final class logits:

$$\mathbf{o}_i = \sigma(\mathbf{W} \cdot \mathbf{s}_i + \mathbf{W}_z \cdot \mathbf{z}_i + \mathbf{b}), \tag{7}$$

where $\mathbf{s}_i \in \mathbb{R}^{N_c}$ is the concept score vector of subject $i$, $\mathbf{W} \in \mathbb{R}^{N \times N_c}$ and $\mathbf{W}_z \in \mathbb{R}^{N \times d}$ are trainable weight matrices, $\mathbf{b} \in \mathbb{R}^N$ is the bias term, $\sigma(\cdot)$ is a nonlinear activation function (*e.g.,* Sigmoid). $d$ is the dimension of embeddings. The final predictive distribution is given by the softmax:

$$p(y \mid i) = \frac{\exp(o_{i,y})}{\sum_{y' \in \mathcal{Y}} \exp(o_{i,y'})}, \qquad \forall y \in \mathcal{Y}. \tag{8}$$

This design ensures that predictions are directly mediated through concept scores, while the MLP provides a flexible mapping from the concept space to the disorder label space.

**Classification loss.** The primary objective is the standard cross-entropy loss over the predicted class distribution: $\mathcal{L}_{\mathrm{cls}} = -\sum_{i=1}^M \log p(y_i \mid i)$, where $p(y \mid i)$ is defined in Eq. (8) through the softmax over class logits.

**Sparsity on concepts.** To encourage the model to rely on a compact set of highly informative concepts, we add an $\ell_1$ penalty on the first-layer weights $\mathbf{W}$ of the MLP, which directly control the influence of each concept on the downstream prediction: $\mathcal{L}_{\mathrm{sp}} = \lambda_{\mathrm{sp}} \|\mathbf{W}\|_1$, where $\lambda_{\mathrm{sp}}$ is a tunable coefficient. This loss promotes sparsity across concepts, making it easier to identify a small subset of clinically meaningful connectivity patterns that drive classification decisions.

**Direction-aware constraints.** If $\delta_c \in \{\pm 1\}$ is provided for a concept $c$, we incorporate this prior into the concept-to-class mapping. Specifically, for concepts labeled with $\delta_c = +1$ (hyperconnectivity), we enforce that their learned contribution to the class logits is non-negative; for $\delta_c = -1$

(hypoconnectivity), the contribution is enforced to be non-positive. Practically, we implement this by adding a hinge penalty that punishes sign violations. This constraint serves as an inductive bias that aligns the learned concept-class associations with domain knowledge.

**Overall objective.** The final training loss combines all terms:

$$\mathcal{L} = -\sum_{i=1}^{M} \log \frac{\exp(o_{i,y_i})}{\sum_{y' \in \mathcal{Y}} \exp(o_{i,y'})} + \lambda_{\text{sp}} \|\mathbf{W}\|_1 + \lambda_{\text{dir}} \sum_{c=1}^{N_c} \sum_{j=1}^{N} \Big[ \max\big(0, -\delta_c \, \mathbf{W}_{j,c}\big) \Big]^2, \quad (9)$$

This formulation ensures that the learned model is both accurate and interpretable, and grounded in both clinical priors and the parsimony of concept usage.

**Summary.** Our framework provides a clinically meaningful bridge between complex fMRI connectivity signals and psychiatric diagnosis by ensuring that predictions are mediated through interpretable connectivity concepts. Unlike black-box models, which often struggle to gain trust in clinical practice, CONCEPTNEURO produces explanations that are directly anchored in neurobiological constructs familiar to clinicians, such as amygdala–prefrontal connectivity or cingulate–insula interactions. This interpretability enables practitioners to not only verify model decisions but also use them as hypotheses for further investigation into disorder mechanisms.

## 4 EXPERIMENTS

We organize our experiments around the following research questions: **RQ1:** Does CONCEPTNEURO consistently outperform vanilla GNN baselines across multiple architectures and disorder prediction tasks? **RQ2:** Do the concepts discovered by CONCEPTNEURO align with domain experts' understanding of clinically meaningful brain connectivity? **RQ3:** What patterns emerge when analyzing the distribution of concept importance across subjects, and do these patterns reflect disorder-specific neural signatures? **RQ4:** How critical are the design choices of CONCEPTNEURO, such as learned concept weights and inclusion of diverse concept sets, for achieving strong predictive performance?

### 4.1 EXPERIMENTAL SETTINGS

**Datasets.** We conduct experiments on two fMRI datasets. (1) Resting-state fMRI (rs-fMRI) data from the Adolescent Brain Cognitive Development (ABCD) Study[1] (Casey et al., 2018), the largest longitudinal neuroimaging study of brain development in youth in the United States. At baseline, 11,099 participants aged 9–11 years were enrolled. After excluding individuals without usable rs-fMRI scans or those who failed the ABCD quality control procedures, our final sample comprised 7,844 participants. All rs-fMRI data were preprocessed using the standard ABCD pipelines (Hagler et al., 2019). Cortical regions were parcellated using the Glasser atlas (Glasser et al., 2016), and subcortical regions were defined using the Aseg atlas (Fischl et al., 2002). Functional connectivity was then estimated as the statistical association between ROI-level rs-fMRI time series. (2) Human Connectome Project-Development (HCP-D) dataset, which contained data from a total of 1300 youth ranging in age from 5 to 21 years (Somerville et al., 2018). In the current release (version 2.0), the baseline data from 652 participants were available. We excluded participants with excessive head motion during scanning, and the final sample included 528 participants (Zhang et al., 2022a). Preprocessed rs-fMRI data using the HCP minimal preprocessing pipelines v3.22 (Glasser et al., 2013) were available. We used the same Glasser atlas for cortical parcellation and applied the same procedure for FC estimation.

**Diagnostic Task.** In this study, we focus on five psychiatric disorders commonly examined in pediatric populations: obsessive-compulsive disorder (OCD), anxiety, attention-deficit/hyperactivity disorder (ADHD), oppositional defiant disorder (ODD), and conduct disorder. Diagnostic information was obtained through the parent-report Kiddie Schedule for Affective Disorders and Schizophrenia for School-Age Children, Computerized Version (K-SADS-COMP) or the Child Behavior Checklist (CBCL) subscales for dimensional assessment (Achenbach & Rescorla, 2001). Particularly, for the ABCD dataset, the task is framed as binary classification, where the goal is to determine whether a subject is diagnosed with a given disorder. In contrast, for the HCP-D dataset, we use the raw symptom score as the label for a multi-class classification task. This symptom score ranges from 0 to

---

[1]https://abcdstudy.org/

Table 1: Mean accuracy (%) on disorder classification tasks on the ABCD dataset, with standard error across five random seeds. We observe that CONCEPTNEURO-augmented GNN architectures consistently outperform their vanilla counterparts across all five classes.

| GNN Architecture | Method | Anxiety | OCD | ADHD | ODD | Conduct |
|---|---|---|---|---|---|---|
| GCN | Vanilla | 61.2±1.7 | 70.1±1.1 | 57.8±1.8 | 65.5±1.3 | 69.3±1.5 |
| | CONCEPTNEURO | **65.8**±1.0 | **76.1**±1.7 | **62.6**±1.3 | **67.5**±1.9 | **72.3**±1.2 |
| GAT | Vanilla | 61.0±1.4 | 69.8±1.2 | 57.2±1.9 | 65.1±1.3 | 68.7±2.0 |
| | CONCEPTNEURO | **64.5**±1.1 | **74.9**±1.5 | **62.0**±1.7 | **66.9**±1.3 | **71.4**±1.6 |
| GraphSAGE | Vanilla | 61.5±1.8 | 70.5±1.3 | 58.1±1.7 | 66.2±2.0 | 69.0±1.4 |
| | CONCEPTNEURO | **65.0**±1.6 | **75.3**±1.2 | **62.2**±1.9 | **67.1**±1.2 | **71.8**±1.7 |
| GIN | Vanilla | 62.7±1.3 | 71.0±1.4 | 61.4±1.1 | 67.9±1.8 | 73.2±1.0 |
| | CONCEPTNEURO | **64.9**±0.9 | **72.5**±1.0 | **63.8**±1.4 | **69.7**±1.2 | **75.9**±0.9 |

10, resulting in an 11-class setup. Importantly, the raw score is not a formal clinical diagnosis; rather, it represents the number of symptoms observed for a specific disorder. As such, it can be interpreted as a proxy for disease severity. For the HCP-D dataset, only four disorders are considered (as the symptom scores of OCD are unavailable).

**Baselines.** To benchmark our proposed framework, we compare it against several widely used models in graph learning and representation learning. Graph Convolutional Network (GCN) (Kipf & Welling, 2017) leverages spectral graph convolutions to aggregate neighborhood information. Graph Isomorphism Network (GIN) (Xu et al., 2019) strengthens expressive power by using injective aggregation functions. Graph Attention Network (GAT) (Veličković et al., 2018) incorporates attention mechanisms to assign learnable weights to neighboring nodes. GraphSAGE (Hamilton et al., 2017) introduces an inductive framework that learns aggregation functions to generalize to unseen nodes and graphs.

**Implementation Details.** The loss weights used in Eq. 9 are set as 1. We use a learning rate of $1 \times 10^{-3}$ and weight decay of $10^{-4}$. We use GPT-4.1 as the LLM for generating concepts. More details are provided in Appendix B.

## 4.2 EXPERIMENTAL RESULTS

Here, we discuss experimental results that answer key questions highlighted at the beginning of this section (RQ1-RQ4).

**RQ1) CONCEPTNEURO outperforms Vanilla Graph Neural Networks.** Table 1 reports the accuracy of different GNN architectures on five disorder classification tasks. We observe several consistent patterns emerge. First, across all four GNN architectures, CONCEPTNEURO substantially outperforms the corresponding vanilla baseline. For example, with GCN, the average improvement ranges from $+2.0\%$ (Conduct) to nearly $+5.0\%$ (Anxiety), with gains consistently larger than the reported standard errors. Similar margins are observed for GAT and GraphSAGE, where our approach improves performance by $3.4\%$ across five disorders. These results highlight the benefit of explicitly incorporating LLM-guided concepts into otherwise standard black-box GNN pipelines.

Second, the improvements are not limited to a single architecture, but hold across convolution-based (GCN), attention-based (GAT), sampling-based (GraphSAGE), and injective (GIN) encoders. This demonstrates that our framework is architecture-agnostic and provides complementary information to the underlying message-passing mechanism. Notably, even strong baselines such as GIN-Vanilla benefit from our design: for ADHD, accuracy increases from $61.4\%$ to $63.8\%$, and for Conduct class from $73.2\%$ to $75.9\%$.

Third, our framework demonstrates strong performance on the multi-class classification task using the HCP-D dataset, as reported in Table 2. Unlike the binary classification setting, here the prediction problem is extended to 11 classes (ranging from 0 to 10), which substantially increases task difficulty

Table 2: Mean accuracy (%) on disorder classification tasks on the HCP-D dataset, with standard error across five random seeds. We observe that CONCEPTNEURO-augmented GNN architectures consistently outperform their vanilla counterparts across all four classes. Note that the HCP-D dataset does not have labels for the OCD class.

| GNN Architecture | Method | Anxiety | ADHD | ODD | Conduct |
|---|---|---|---|---|---|
| GCN | Vanilla | 29.9±2.1 | 45.8±1.9 | 34.1±1.6 | 19.8±1.5 |
| | CONCEPTNEURO | **34.8**±2.5 | **52.8**±2.0 | **40.2**±2.4 | **28.7**±2.3 |
| GAT | Vanilla | 28.1±1.5 | 44.0±2.7 | 35.0±2.8 | 18.6±2.6 |
| | CONCEPTNEURO | **34.8**±2.3 | **53.2**±2.4 | **44.8**±2.5 | **31.0**±1.5 |
| GraphSAGE | Vanilla | 29.4±1.6 | 45.8±1.8 | 37.6±2.9 | 22.7±2.4 |
| | CONCEPTNEURO | **38.3**±2.2 | **54.3**±2.5 | **46.9**±1.6 | **30.9**±2.1 |
| GIN | Vanilla | 31.1±1.4 | 48.7±1.6 | 35.4±2.5 | 21.3±2.3 |
| | CONCEPTNEURO | **38.6**±2.4 | **56.4**±1.8 | **47.1**±2.4 | **30.3**±2.1 |

and reduces the baseline accuracy across all architectures. Despite this, CONCEPTNEURO consistently outperforms the vanilla GNN counterparts across all four disorders.

Our results reveal that different disorders vary in difficulty. While tasks such as ADHD and Anxiety yield lower absolute accuracies across all methods, reflecting their more heterogeneous neural signatures, *Conduct* disorder consistently achieves higher accuracy. Importantly, in all cases, our method narrows this performance gap while retaining interpretability, suggesting that concept-based modeling helps extract clinically meaningful connectivity patterns that align with diagnostic labels.

Overall, our findings show that CONCEPTNEURO consistently improves predictive performance across diverse GNN backbones, while maintaining generalizability to multiple disorder types.

Table 3: Agreement between model-extracted concepts and expert-selected concepts. We report two metrics: (i) **Concept Agreement**, the fraction of shared concepts at top–$k$, and (ii) **Ranking Agreement**, the ratio of subjects for which the model-selected features matched expert-selected features. Both are shown at top–3, top–5, and top–10.

| | Concept Agreement | | | Ranking Agreement | | |
|---|---|---|---|---|---|---|
| **Disorder** | **Top–3** | **Top–5** | **Top–10** | **Top–3** | **Top–5** | **Top–10** |
| Anxiety | 66.7% | 80.0% | 70.0% | 61.2% | 73.5% | 81.0% |
| OCD | 33.3% | 40.0% | 80.0% | 47.6% | 59.2% | 77.8% |

**RQ2) Domain Expert Analysis/Verification.** To further validate the interpretability of our framework, we compared the model-extracted concepts (ranked by concept scores in Eq. 6) with those selected by clinical domain experts in neuroimaging and psychiatry. Agreement was assessed using two complementary metrics: (i) *Concept Agreement*, which measures the proportion of shared concepts between the top-$k$ sets from the model and experts, and (ii) *Ranking Agreement*, which quantifies the average per-subject alignment of selected features across the cohort.

As shown in Table 3, concept agreement values are consistently strong at larger cutoffs (*e.g.,* 70–80% at top–10), suggesting that our framework captures many clinically meaningful connectivity patterns. At stricter cutoffs, concept agreement is more variable across disorders (*e.g.,* 66.7% for Anxiety vs. 33.3% for OCD at top–3), reflecting differences in the prioritization of the most salient features. The ranking agreement metric further supports these findings: for both Anxiety and OCD, agreement rates increase steadily from top–3 ($\approx$47–61%) to top–10 ($\approx$78–81%), demonstrating that our framework recovers clinically validated concepts consistently across subjects.

Overall, these results demonstrate that CONCEPTNEURO not only provides competitive predictive performance but also yields interpretable concept-level explanations that align well with domain expertise, both at the set-level (via ranking overlap) and subject-level (via ranking agreement).

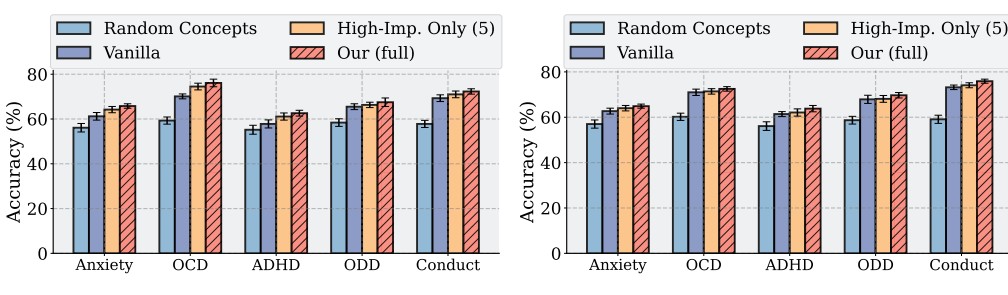

(a) GCN ablation results across disorders.          (b) GIN ablation results across disorders.

Figure 3: Ablation study results for two architectures.

**RQ3) Concept Analysis of Importance Distributions over Subjects.** To further investigate how individual concepts contribute to disorder prediction, we focus on the case of *Anxiety* and select two representative concepts: the most important concept (*Altered connectivity from orbitofrontal cortex (OFC) to amygdala*) and the least important concept (*Increased connectivity between thalamus and prefrontal regions*). For each of these concepts, we compute the distribution of cosine similarity scores across all subjects, as shown in Fig. 2. The results reveal clear differences in their subject-level relevance.

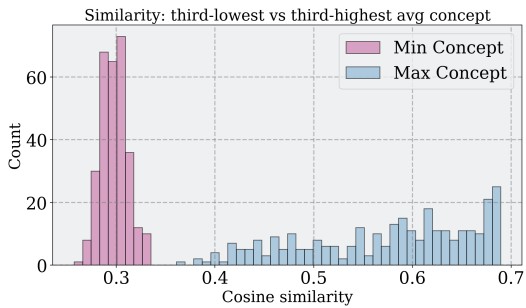

Figure 2: The distribution of similarity scores.

The most important concept consistently aligns with a large subset of subjects and represents a core connectivity pattern strongly associated with Anxiety. In contrast, the least important concept displays a distribution centered closer to zero, suggesting weak relevance across subjects. This comparison highlights how our framework quantifies relevance of concepts across individuals. By examining these distributions, clinicians can distinguish between robust, disorder-specific connectivity markers and marginal features, thereby improving the interpretability and reliability of model outputs.

**RQ4) Ablation Study.** To disentangle the contribution of each design choice in CONCEPTNEURO, we perform two ablation variants: (1) replacing our achieved concepts with randomly sampled concepts, and (2) restricting the model to use only the top-5 most important concepts. The results are shown in Fig. 3. First, the **random concepts** setting results in a substantial drop in accuracy across all disorders and architectures. This degradation confirms that the functional connectivity concepts extracted by CONCEPTNEURO are not arbitrary; rather, they encode clinically meaningful neurobiological patterns. In particular, models trained with random concepts perform close to chance level, highlighting the necessity of guided concept generation. Second, the **top–5 concepts only** setting performs better than random concepts and even approaches the full model in some cases, especially for disorders like Anxiety and OCD where a small subset of connectivity patterns are highly discriminative. However, across all tasks the top–5 setting still underperforms the full model, demonstrating that while a few high-importance concepts are valuable, broader concept coverage is essential for capturing the heterogeneity of psychiatric disorders. Overall, these ablations show that both the diversity and quality of the extracted concepts are critical. The full CONCEPTNEURO framework thus provides the best balance between interpretability and predictive performance.

## 5  CONCLUSION

In this work, we introduced an interpretable framework for neuropsychiatric diagnosis that combines large language models (LLMs) and graph neural networks (GNNs) through the use of connectivity-based concepts. By leveraging LLMs to generate clinically grounded candidate concepts, our method enables predictions that are both accurate and interpretable. The concept bottleneck classifier enforces sparsity and direction-aware regularization, ensuring that model decisions align with clinical priors while relying on a compact set of meaningful connectivity patterns. Extensive experiments across multiple disorders demonstrated that our approach achieves strong predictive performance, while expert analysis verified that the identified concepts capture clinically relevant connectivity features.

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

CONTENTS OF THE APPENDIX

## A    LIMITATIONS STATEMENT

While our framework advances interpretable neuropsychiatric diagnosis by integrating LLM-generated connectivity concepts with concept-guided GNNs, several limitations remain. First, our evaluation is conducted on publicly available datasets (e.g., ABCD, HCP-D), which, although widely used, may contain biases in demographics, acquisition protocols, and diagnostic labels. These factors could limit generalizability to other populations or clinical settings. Second, the interpretability of our method is inherently constrained by the quality of the candidate concepts provided by LLMs; although we employ filtering and expert validation, spurious or incomplete concepts may still arise. Third, while our direction-aware regularization incorporates clinical priors, it assumes that such priors are correct and consistent across individuals, which may not always hold in heterogeneous disorders. Finally, our framework is intended as a research tool and has not been clinically validated. Translation to practice would require large-scale prospective studies, integration with multimodal assessments, and careful oversight to ensure safety, fairness, and reliability.

## B    EXPERIMENTAL SETTINGS

We employ a 2-layer GNN with hidden dimension of 64, where each layer is followed by BatchNorm and ReLU activation, and global mean pooling is used to obtain graph embeddings. Regularization is applied through node dropout with probability $0.5$ and weight decay of $10^{-4}$. The Adam optimizer is used with a learning rate of $1 \times 10^{-3}$. Training is performed for $500$ epochs with $100$ balanced mini-batches per epoch, each containing up to 16 positive and 16 negative samples. Validation is conducted every 5 epochs, and early stopping patience is set to 20. All experiments are conducted on a NVIDIA A6000 GPU with 48GB of memory.

## C    GENERATED CONCEPTS (UNFILTERED)

| Concepts for Anxiety |
|---|
| • hyperconnectivity between amygdala and prefrontal cortex |
| • aberrant connectivity between parahippocampal gyrus and cingulate cortex |
| • altered connectivity from orbitofrontal cortex (OFC) to amygdala |
| • decreased connectivity between dorsolateral prefrontal cortex and orbitofrontal cortex |
| • decreased connectivity between occipital cortex and prefrontal cortex |
| • decreased connectivity between right hippocampus and precuneus |
| • dysfunctional communication between cingulate cortex and frontal regions |
| • dysregulated connectivity within anterior cingulate cortex (ACC) |
| • elevated connectivity between DLPFC and cingulate cortex |
| • elevated connectivity between insula and amygdala |
| • elevated insula to somatosensory cortex connectivity |
| • enhanced functional coupling between parahippocampal gyrus and occipital cortex |
| • heightened synchronization between amygdala and hippocampus |
| • hyperconnectivity within anterior cingulate cortex |
| • altered connectivity between left hippocampus and prefrontal cortex |
| • increased connectivity between frontal cortex and brainstem |
| • increased connectivity between right hippocampus and prefrontal regions |
| • increased connectivity between thalamus and prefrontal regions |
| • increased connectivity in parahippocampal-prefrontal networks |
| • increased functional connectivity between insula and anterior cingulate cortex |
| • reduced functional connectivity between DLPFC and limbic regions |

**Concepts for ODD**

- altered functional connectivity between the right prefrontal cortex and striatum
- altered functional connectivity between the left prefrontal cortex and the striatum
- aberrant connectivity between the ventral striatum and precuneus
- abnormal connectivity patterns in the inferior frontal gyrus
- altered functional coupling between the cerebellum and prefrontal cortex
- attenuated connectivity between the left inferior frontal gyrus and occipital regions
- augmented connectivity between superior frontal areas and motor cortex
- decreased functional connectivity between the motor cortex and reward regions
- diminished functional connectivity from the superior frontal gyrus to the precuneus
- dysregulated connectivity from the ventral striatum to the left motor cortex
- elevated functional connectivity in the left orbitofrontal cortex
- enhanced connectivity between the cingulate cortex and somatomotor areas
- higher functional connection between the left inferior frontal gyrus and the right striatum
- hyperconnectivity of the insula with the anterior cingulate cortex
- hyperconnectivity within the right orbitofrontal cortex
- impaired connectivity between the prefrontal cortex and somatosensory areas
- increased synchronization in the superior frontal gyrus networks
- increased synchrony between the cerebellum and occipital regions
- lowered connectivity between the orbitofrontal cortex and cingulate cortex
- reduced connectivity between the dorsolateral prefrontal cortex and occipital cortex
- reduced functional connectivity within the right insula

**Concepts for OCD**

- altered functional connectivity between orbitofrontal cortex and right thalamus
- abnormal connectivity between insula and frontal lobe
- abnormal connectivity between the occipital lobe and anterior cingulate
- abnormal functional connections between right insula and anterior cingulate cortex
- abnormal functional connectivity between the right basal ganglia and frontal lobe
- altered connectivity between the cerebellum and prefrontal cortex
- altered connectivity between the right OFC and left thalamus
- altered functional connectivity in the basal ganglia
- elevated connectivity between the caudate nucleus and prefrontal cortex
- elevated connectivity between the left orbitofrontal cortex and right anterior cingulate
- enhanced connectivity between the striatum and prefrontal cortex
- hyperconnectivity between left orbitofrontal cortex and basal ganglia
- hyperconnectivity between the anterior cingulate cortex and striatum
- hyperconnectivity between the left orbitofrontal cortex and left anterior cingulate
- hyperconnectivity between the superior temporal gyrus and cingulate
- hyperconnectivity in the anterior cingulate cortex
- increased functional connectivity between the right superior frontal gyrus and thalamus
- reduced functional connectivity between occipital lobe and frontal areas
- altered connectivity between the left OFC and left thalamus

| Concepts for ADHD |
| --- |
| • decreased functional connectivity between prefrontal cortex and striatum |
| • abnormal FC between orbitofrontal cortex and thalamus |
| • altered connectivity between anterior cingulate cortex and cerebellum |
| • altered connectivity between caudate and sensorimotor cortex |
| • altered connectivity between cingulate and sensorimotor cortex |
| • attenuated connectivity between left orbitofrontal cortex and motor cortex |
| • attenuated connectivity between superior frontal gyrus and occipital lobe |
| • attenuated functional connectivity within the working memory network |
| • decreased connectivity between caudate and precuneus |
| • decreased connectivity between inferior frontal gyrus and insula |
| • decreased connectivity between the cerebellum and motor cortex |
| • decreased connectivity within the default mode network (DMN) |
| • decreased functional connectivity between precuneus and occipital lobe |
| • diminished task-related functional connectivity in the right frontal lobe |
| • hypo-connectivity between cingulate cortex and frontal regions |
| • hypo-connectivity within reward processing networks |
| • hypoconnectivity between the cerebellum and prefrontal regions |
| • impaired connectivity between right inferior frontal gyrus and superior frontal gyrus |
| • lower connectivity between the default mode network and superior frontal gyrus |
| • lower synchrony between occipital lobe and reward network |
| • lowered connectivity between motor cortex and somatosensory regions |
| • reduced connectivity between orbitofrontal cortex and insula |
| • reduced connectivity between working memory areas and anterior insula |
| • reduced functional connectivity within the anterior cingulate cortex |
| • reduced FC between insula and sensorimotor network |
| • reduced integration between dorsolateral prefrontal cortex and precuneus |
| • weakened connectivity between caudate and anterior cingulate cortex |
| • weaker connectivity between right inferior frontal gyrus and thalamus |

> **Concepts for Conduct Disorder**
>
> - reduced functional connectivity between anterior cingulate cortex and prefrontal cortex
> - reduced functional connectivity between the amygdala and prefrontal cortex
> - abnormal connectivity between fusiform gyrus and occipital cortex
> - abnormal functional connectivity between the cerebellum and prefrontal cortex
> - altered connectivity between the left superior frontal gyrus and default mode network
> - altered functional connectivity between brainstem and prefrontal cortex
> - altered functional connectivity within the default mode network
> - attenuated connectivity between orbitofrontal cortex and anterior cingulate cortex
> - attenuated thalamic connectivity with prefrontal regions
> - decreased connectivity between amygdala and orbitofrontal cortex
> - decreased connectivity between fusiform gyrus and anterior cingulate cortex
> - decreased connectivity between the right superior frontal cortex and motor areas
> - diminished connectivity between precuneus and superior frontal gyrus
> - diminished connectivity between right superior frontal gyrus and default mode network
> - enhanced connectivity within the insula
> - hyperconnectivity between the insula and anterior cingulate cortex
> - hypoconnectivity between the anterior brainstem and cingulate cortex
> - impaired functional links between anterior cingulate cortex and supplementary motor area
> - reduced connectivity between cingulate cortex and cerebellum

# D  DISORDER-RELATED TERMS

To provide additional guidance for concept generation, we extract disorder-related terms from NeuroQuery (Dockès et al., 2020). These terms serve as anchors to help ensure that the generated concepts are clinically meaningful and aligned with prior neuroimaging knowledge.

- **Anxiety Disorders:** amygdala, prefrontal, insula, cingulate, thalamus, occipital, brainstem, somatization, dlpfc, orbitofrontal, left, acc, right, hippocampus, fa, fusiform gyrus, memory, precuneus, parahippocampal, motion, ofc, hope, frontal, task
- **Oppositional Defiant Disorder (ODD):** reward, fa, cerebellum, white matter, tensor, matter, occipital, precuneus, white, cingulate, prefrontal, insula, frontal, motion, striatum, superior, right, motor, inferior, left, orbitofrontal, task, striatal, somatization
- **Attention-Deficit/Hyperactivity Disorder (ADHD):** cerebellum, precuneus, cingulate, occipital, insula, prefrontal, frontal, motor, inferior, right, striatal, superior, default, left, orbitofrontal, motion, task, reward, acc, sensorimotor, caudate, lobe, working memory, thalamus
- **Obsessive-Compulsive Disorder (OCD):** thalamus, cingulate, frontal, ofc, fa, occipital, insula, striatal, white matter, right, left, cerebellum, acc, anterior, orbitofrontal, tensor, stg, nucleus, superior, task, lobe, matter, basal ganglia
- **Conduct Disorder:** insula, dmn, fusiform, matter, occipital, precuneus, fusiform gyrus, thalamus, prefrontal, cingulate, sma, orbitofrontal, motion, amygdala, ofc, frontal, anterior, brainstem, cerebellum, default, right, superior, motor, default mode

# E  REPRODUCIBILITY STATEMENT

We have taken careful steps to ensure that our framework and results are reproducible. The entire codebase, including data preprocessing scripts, model implementation, and training procedures, is provided in our **anonymous repository**: https://anonymous.4open.science/status/ConceptNeuro. Exact **hyperparameters**, training configurations, and optimizer settings are reported in Appendix B, along with information about early stopping and regularization. We document all **datasets, preprocessing pipelines, and region atlases** used in our experiments in Sec. 4.1. To facilitate verification,

we include **ablation studies**, **expert analysis results**, and **visualizations** (Tables/Figures in the main text and appendices). The repository also contains environment files, data loaders, and evaluation scripts to guarantee that all reported results can be replicated.

## F   THE USE OF LARGE LANGUAGE MODELS

According to the ICLR's policy on the use of large language models (LLMs), we explicitly state how LLMs were employed in this work. Our research investigates how LLMs can be leveraged to generate clinically meaningful concepts for neuropsychiatric diagnosis. LLMs were directly integrated into the methodology by serving as a concept generation module, prompted with structured queries to produce disorder-related functional connectivity concepts, which were subsequently filtered and integrated into our framework. Beyond this methodological role, LLMs were also used as auxiliary tools to polish the manuscript's presentation by improving grammar and readability. Importantly, all core scientific contributions, including the design of algorithms, experimental implementation, and analyses, were conceived and validated entirely by the authors.

## G   ETHICS STATEMENT

This research does not involve the collection or use of personal, sensitive, or identifiable data. All experiments are conducted on publicly available neuroimaging datasets, such as ABCD and HCP-D, which are widely used in the neuroscience and machine learning communities. These datasets are de-identified and shared under strict data usage agreements, ensuring compliance with ethical standards for human subject research.

While our framework is designed to enhance interpretability and align predictions with clinical priors, we acknowledge that automated tools in psychiatry and neuroscience should be deployed with caution. In particular, care must be taken to avoid over-reliance on machine predictions, to ensure human oversight in clinical decision-making, and to mitigate risks such as model bias, misinterpretation of neuroimaging findings, or unintended misuse in sensitive healthcare contexts. The methods and results presented in this paper are intended strictly for research purposes, and any potential translation to clinical practice must be accompanied by rigorous validation and ethical review.

