# OpenReview forum: "Interpretable Neuropsychiatric Diagnosis via Concept-Guided Graph Neural Networks"
_ICLR.cc/2026/Conference — ICLR 2026 Conference Withdrawn Submission_

### Official Review · Reviewer_2ECq · 2025-10-22

**Soundness:** 2
**Presentation:** 3
**Contribution:** 2
**Rating:** 2
**Confidence:** 4

**Summary:**

This study proposes an interpretable GNN-based architecture that integrates domain concepts generated by GPT-4.1, mainly aimed at neurodevelopmental diagnosis. Compared to the vanilla GNN, the model demonstrates improved predictive performance. However, the interpretability evidence presented in the paper appears limited and somewhat insufficient to substantiate the claimed clinical insights.

**Strengths:**

1. The study employs LLM to generate cortical concepts of interest, which are then used to extract both whole-brain and subgraph functional connectivity (FC) features, achieving a local–global representation.
2. The proposed model is lightweight, consisting of only a two-layer GNN.

**Weaknesses:**

1. Although the paper emphasizes internal interpretability and clinical meaning, there are very few explicitly interpretable results presented. Apart from the example shown in line 448, the interpretability claims remain largely unsubstantiated. I am curious about whether the identified FCs truly correspond to disease-specific patterns.
2. Regarding the evaluation of explanation quality, it remains unclear how the “gold standard” concepts from domain experts were obtained. How reliable and consistent are these expert-provided references?
3. In my view, many of the unfiltered concepts listed in Appendix C could also be derived from meta-analyses or review articles. What advantages does the proposed LLM-based approach offer compared to such traditional methods? Additionally, have the authors considered employing RAG to better ground LLM outputs in neuroscience literature?
4. I was unable to find usable code or implementation details in the provided anonymous repository, despite the authors’ statement (line 914):
>  “The entire codebase, including data preprocessing scripts, model implementation, and training procedures, is provided in our anonymous repository: https://anonymous.4open.science/status/ConceptNeuro ”

**Questions:**

1. I am curious about the selection of the hyperparameter $N_c$ , i.e., the number of concepts. How was this value determined? It appears to play an important role, as suggested by the top-5 concept in the ablation studies.
2. The unfiltered concept pool contains around 20 concepts per disease. Given the concept filtering process and the sparse loss term, I tend to believe that the model may ultimately select fewer than five dominant concepts, even though it dynamically selects top-k concepts. Could the authors clarify how many concepts are effectively retained after training (sparsity)?
3. I assume that the authors split the ABCD dataset into five subsets for evaluation. Could the authors clarify the number of patients and healthy controls included in each subset?

---

### Official Review · Reviewer_1iYc · 2025-10-25

**Soundness:** 2
**Presentation:** 2
**Contribution:** 2
**Rating:** 4
**Confidence:** 4

**Summary:**

- Proposes a concept‑guided GNN framework with an explicit concept bottleneck that links interpretable functional‑connectivity subgraph concepts to diagnostic predictions on rs‑fMRI for neuropsychiatric disorders.
- Describes an LLM‑based pipeline to automatically generate, filter, and structurally encode disorder‑related functional‑connectivity concepts grounded in neuroimaging knowledge resources.
- Reports empirical results across multiple datasets and GNN backbones, showing consistent improvements over vanilla counterparts, including in multi‑class severity prediction settings.
- Provides concept‑level analyses (e.g., importance distributions and case visualizations) indicating alignment with established neurobiological priors to support interpretability claims.

**Strengths:**

## Strengths
- Methodological novelty: introduces a concept‑bottleneck, aligning interpretable functional‑connectivity subgraph concepts with predictions. Compared to standard GNN pipelines, the decision pathway is clearer and more auditable, providing an interpretable rs‑fMRI diagnostic route that can facilitate clinical communication and hypothesis generation.
- Knowledge injection: leverages LLMs together with neuroimaging resources to automatically generate and filter concepts, yielding a systematic and extensible concept library.
- Strong interpretability: concepts are defined as structured subgraphs; the paper reports subject‑level Top‑K concepts and population‑level distributions, supported by expert assessment/selection.
- Broad empirical coverage: consistent gains over vanilla counterparts across multiple datasets and GNN backbones, with validation on multi‑class severity prediction tasks.
- Thorough ablations: replacing concepts with random ones or restricting to a very small set substantially degrades performance, underscoring the critical role of concept quality and supporting the methodological claim.

## Novelty and Significance
- The paper combines LLM‑driven knowledge injection with an explicit concept bottleneck in GNNs for rs‑fMRI diagnosis, yielding a structured and interpretable concept layer; robustness is demonstrated across multiple GNN backbones.
- Compared to CBM/prototype approaches and post‑hoc explainers, the work emphasizes a forward, prior‑to‑prediction interpretability pathway (“prior concepts → structured subgraph → prediction”).
- Significance: establishes a clinically communicable route for graph learning on rs‑fMRI, with potential for cross‑task transfer and hypothesis generation in neuroimaging.

**Weaknesses:**

## Weaknesses
- Robustness and reproducibility of LLM‑driven concept generation are under‑validated (sensitivity to prompts, LLM variants, random seeds). The study appears to rely on GPT‑4.1; please add sensitivity analyses to alternative models/prompts.
- Limited comparisons against strong interpretability baselines (e.g., GNNExplainer, PGExplainer, prototype/CBM variants), which weakens the positioning of the method.
- Prompting is under‑specified: provide complete prompt templates and construction rationale, assess prompt robustness, and release prompts to improve reproducibility.
- Direction‑aware constraints are insufficiently described; include ablations isolating the effect/necessity of the direction‑aware loss term (with formulation and hyperparameters) and discuss observed behaviors.

## Clarity and Presentation
- The manuscript is well structured and reads clearly. To further improve clarity and reproducibility, I recommend:
  - Providing the complete LLM-based concept generation prompt templates, together with representative examples of filtered concepts and illustrative failure cases;
  - Completing the anonymized code repository and including scripts and documentation for one-click reproduction of experiments and figures.

**Questions:**

- Section 3.1 (around line 153): As I understand it, should $V_c^{1}$ and $V_c^{2}$ each be subsets of $V$, rather than $(V_c^{1}, V_c^{2})$ being a subset of $V$? Could you clarify the intended set relationship and revise the notation accordingly?
- Section 3.3 (around line 215): Could you provide a brief description of NetworkX usage, algorithm variants and hyperparameters, and a short rationale for how these choices affect concept subgraph construction and results?
- Around line 218: You state “Formally, the node representation at layer $t$ is updated as:”, but Eq. (3) uses $h_u^{(0)}$ with a superscript 0—does this refer to a layer-$t$ update or a layer-0 initialization? If it is a typographical error, could you correct the equation and adjust subsequent derivations/results as needed?

---

### Official Review · Reviewer_qWR3 · 2025-10-28

**Soundness:** 2
**Presentation:** 2
**Contribution:** 2
**Rating:** 2
**Confidence:** 5

**Summary:**

This work presents an interpretable framework for neuropsychiatric diagnosis that integrates large language models (LLMs) and graph neural networks (GNNs) using connectivity-based concepts. The proposed method utilizes LLMs to generate clinically relevant candidate concepts, enabling interpretable predictions. The concept bottleneck classifier incorporates sparsity and direction-aware regularization, ensuring that model decisions align with clinical knowledge while focusing on a compact set of connectivity patterns.

**Strengths:**

1. Leverages prior knowledge from LLMs to generate clinically relevant candidate concepts.

2. The writing of the paper is clear, and the experimental results appear promising.

**Weaknesses:**

1. The proposed method heavily relies on the prior knowledge of LLMs in clinical medicine. If the LLM outputs contain subtle hallucinations or errors, it could significantly affect both the model's performance and its clinical applicability.

2. Pearson's correlation is used to calculate the relationship weights between ROIs. Can this linear approach truly capture the complex relationships between brain regions? Previous works have highlighted this issue and employed attention mechanisms to construct brain maps [1]. Could a deeper analysis and comparison be provided?

3. The comparison methods used are outdated, primarily from 2017–2019. Could experiments be added that compare with more recent methods from the past three years?

4. The paper claims that the method is interpretable. Can meaningful clinical biomarkers be provided through the method and experiments?

5. The proposed method mainly introduces concepts and improves upon the GNN architecture. Can the authors further elaborate on the technical innovations in their approach?

[1] Wang, Yibin, et al. "Residual graph transformer for autism spectrum disorder prediction." Computer Methods and Programs in Biomedicine 247 (2024): 108065.

**Questions:**

Please see Weakness.

---

### Official Review · Reviewer_eWaW · 2025-10-31

**Soundness:** 3
**Presentation:** 2
**Contribution:** 3
**Rating:** 4
**Confidence:** 4

**Summary:**

The author provides a framework for performing neuropscychiatric diagnosis using interpretable connectivity concepts. Specifically, the autfirst prompt LLMs to generate disorderspecific functional connectivity concepts. Then, for each subject, the framework encodes the concepts along with the subject’s input functional connectivity graph. The embeddings are then used to predict a concept score and predictions for downstream tasks. The authors ran extensive experiments to compare the model's performance with vanilla baselines. Ablations are also performed to compare the impact of concept on model's performance.

**Strengths:**

- I like the idea of LLM generated concepts and using these concepts to guide model training. LLM encompasses useful priors as it is trained extensively on medical context. Leveraging these priors provides a novel way to both improve the performance and provide more insights that are aligned with the clinical knowledge.

- The method is model agnostic. The author tested the model against multiple GNN baselines and the model showed statistically significant improvements on all models.

**Weaknesses:**

- While the author tested extensive baselines, all baselines are relatively old (before 2020). It would be nice if the authors can test some newer models, like https://www.sciencedirect.com/science/article/abs/pii/S1361841521002784, https://arxiv.org/abs/2204.07054 or https://arxiv.org/abs/2210.06681. The results would be more convincing if the method also performs well on these newer datasets.

- With the LLM generated insights, it would be nice if the authors can identify and share the most useful concept for each disease / condition. These can be used as useful clinical insights that can be used to guide further research in the area.

- Table 3 shows interesting numbers about concept agreements, but no baselines are compared. For GNN posthoc explanation methods (i.e. https://link.springer.com/chapter/10.1007/978-3-031-16452-1_36), the authors can also potentially identify the important connections among samples, and evaluate their agreement scores. It would be nice if some baseline comparisons are included.

**Questions:**

Please see weaknesses. In particular, I would be willing to raise my score if all my concerns in the weaknesses section are addressed.

---

### Note · Authors · 2025-12-11

I have read and agree with the venue's withdrawal policy on behalf of myself and my co-authors.